# Effects of side-effect risk framing strategies on COVID-19 vaccine intentions: a randomized controlled trial

**Nikkil Sudharsanan[1,2]\*, Caterina Favaretti[1,2], Violetta Hachaturyan[2], Till Bärnighausen[2], Alain Vandormael[2]**

[1]Assistant Professorship of Behavioral Science for Disease Prevention and Health Care,Technical University of Munich, Munich, Germany; [2]Heidelberg Institute of Global Health, Heidelberg University, Heidelberg, Germany

## Abstract

**Background:** Fear over side-effects is one of the main drivers of COVID-19 vaccine hesitancy. A large literature in the behavioral and communication sciences finds that how risks are framed and presented to individuals affects their judgments of its severity. However, it remains unknown whether such framing changes can affect COVID-19 vaccine behavior and be deployed as policy solutions to reduce hesitancy.

**Methods:** We conducted a pre-registered randomized controlled trial among 8998 participants in the United States and the United Kingdom to examine the effects of different ways of framing and presenting vaccine side-effects on individuals' willingness to get vaccinated and their perceptions of vaccine safety.

**Results:** Adding a descriptive risk label ('very low risk') next to the numerical side-effect and providing a comparison to motor-vehicle mortality increased participants' willingness to take the COVID-19 vaccine by 3.0 percentage points (p=0.003) and 2.4 percentage points (p=0.049), respectively. These effects were independent and additive and combining both framing strategies increased willingness to receive the vaccine by 6.1 percentage points (p<0.001). Mechanistically, we find evidence that these framing effects operate by increasing individuals' perceptions of how safe the vaccine is.

**Conclusions:** Low-cost side-effect framing strategies can meaningfully affect vaccine intentions at a population level.

**Funding:** Heidelberg Institute of Global Health.

**Clinical trial number:** German Clinical Trials Registry (#DRKS00025551).

**\*For correspondence:**
nikkil.sudharsanan@tum.de

## Editor's evaluation

This timely online randomized clinical trial is based on 8,998 participants from the U.S. and the U.K. to examine the association between risk-framing nudges and the willingness to get a Covid vaccine. This manuscript would be of interest to behavioral scientists, particularly behavioral economists. Findings from this work indicate that framing matters and can substantially increase the willingness to get vaccinated.

## Introduction

Vaccination is one of the main strategies for controlling the COVID-19 pandemic. However, vaccination rates have slowed and are far from target levels in countries like the United States and the United

**eLife digest** Vaccination is one of the main strategies for controlling the COVID-19 pandemic. But vaccination rates have slowed and are below target levels in countries like the United States and the United Kingdom. While there are many causes of vaccine hesitancy, several studies have found that fear of side effects is the one of the most important.

Although COVID-19 vaccine side-effects are rare, how the media presents these risks may amplify concerns. Addressing public concerns over vaccine side effects is key to improving the uptake of vaccines and booster doses, which has been even lower than primary vaccine series uptake.

Studies show that how risk is presented affects people's risk perceptions and behavior. To learn more about how COVID-19 vaccine risk framing affects risk perception, Sudharsanan et al. enrolled 8,998 people from the United States and the United Kingdom in an online randomized controlled trial. Participants received information about a hypothetical new COVID-19 vaccine, including its side effect rate, and reported their perception of safety and whether they would take the vaccine.

The experiments showed that adding the label "very low risk" when describing vaccine side effect rates increased the number of people who said they would take the vaccine by three percentage points. Comparing the risks of the hypothetical vaccine to the much higher chances of motor vehicle deaths increased an individual's willingness to take the vaccine by 2.4 percentage points. Combining both framing strategies increased people's desire to get vaccinated by 6.1 percentage points.

Deploying these two strategies in vaccine risk communications may help increase primary and booster vaccinations against COVID-19. A next step would be to measure both vaccination intentions and vaccination rates to confirm these strategies.

Kingdom. For example, in the United States, the share of the population that is fully vaccinated went from 2.4% in February 2021 to 34.5% in May 2021 but has since only risen to 63.9% in the following 9 months (*Ritchie et al., 2022*). While not as stagnant, there is a similar pattern in the United Kingdom, where the share of the population that is fully vaccinated was just 71.3% as of February 9, 2022 (*Ritchie et al., 2022*). Despite current recommendations that all adults take a third vaccine dose, rates of boosting are even lower. As of February 9, 2022, just 27.2% of adults in the United States and 55.1% of adults in the United Kingdom have received a booster; importantly, even among those that received two doses, only 42.4% in the United States and 77.3% in the United Kingdom have received a third (*Centers for Disease Control and Prevention, 2021a*; *UK Health Security Agency, 2022*). These vaccination trends have been insufficient to prevent the spread of COVID-19, especially the Omicron variant, which has re-ignited the pandemic in both countries (*Ritchie et al., 2022*; *Centers for Disease Control and Prevention, 2021a*; *UK Health Security Agency, 2022*; *UK Health Security Agency, 2021*; *Aw et al., 2021*).

Vaccine hesitancy is not the result of a single homogenous cause and can vary for different individuals and population groups. For example, recent studies have identified several potential reasons for COVID-19 vaccine hesitancy, including a low perceived risk of COVID-19 infection and concern around how quickly the vaccines were developed (*Aw et al., 2021*; *Robertson et al., 2021*; *Griffith et al., 2021*; *Machingaidze and Wiysonge, 2021*; *Steinert et al., 2021*). A common finding across these studies is that fear and concern about vaccine side-effects is an important reason for vaccine hesitancy. These concerns were potentially heightened by widespread media coverage of vaccine side-effects in April and May 2021, along with the pausing of vaccination efforts in several countries due to this media coverage (*Davey, 2021*; *Mueller and Eddy, 2021*). Although COVID-19 vaccine side-effects rates are extremely low (1 per 100,000 people vaccinated with AstraZeneca in the European Union) (*World Health Organization, 2021*), these rates were often presented by the media without context (*Mueller and Eddy, 2021*; *Karp, 2021*; *Rivas, 2021*; *Saunt, 2021*), likely leading some individuals to reject vaccine uptake (*Pinna et al., 2021*). In addition, the pausing of global COVID-19 vaccination efforts may have sent a strong signal that side-effects are a major cause of concern, thus increasing vaccine hesitancy.

Addressing public concerns over vaccine side-effects will be a key component of efforts to improve vaccine use in the United States, United Kingdom, and globally – especially with recommendations for ongoing booster doses. Several recent behavioral sciences studies investigating this topic have found

that small financial incentives and reminders can improve COVID-19 and flu vaccine uptake (*Campos-Mercade et al., 2021*; *Milkman et al., 2021b*; *Milkman et al., 2021a*; *Dai et al., 2021*; *Klüver et al., 2021*; *Santos et al., 2021*). As another distinct 'nudge' pathway, there is a large body of evidence in health communication and the behavioral sciences that shows that how risks are framed and presented to individuals can also affect their perceptions of its severity and ultimately their behavior (*Bonner et al., 2021*; *Trevena et al., 2013*; *Zipkin et al., 2014*). For example, studies have shown that whether numerical risks are presented as percentages or natural frequencies (e.g. 1% compared to 1 out of 100), with a comparison to a commonly understood but different risk (e.g. the risk of motor-vehicle mortality), using infographics (e.g. visually showing the numbers of individuals in the numerator and denominator of a risk), with descriptive labels (e.g. putting 'very low risk' or 'high risk' next to the numerical risk) can all influence behavior. Such nudges are viewed positively by policymakers since they are often affordable to implement and have the potential for wide population reach (*Thaler and Sunstein, 2009*).

These types of framing effects do not affect behavior by changing deeply held attitudes and beliefs, or by creating strong incentives for a certain behavior. Rather, research has shown that individuals have a limited cognitive ability to process and internalize risk information, especially rare risks like COVID-19 side-effects, and that individuals use mental guides or 'heuristics' to make sense of risk information that ultimately guides their decisions (*Tversky and Kahneman, 1974*; *Gigerenzer and Gaissmaier, 2011*; *Trumbo, 2002*). Therefore, framing effects work by modifying the heuristics that individuals use to understand a risk in a way that does not strongly affect their incentives or remove their agency.

What remains unknown is the impact of these framing effects on COVID-19 vaccine intentions, which framing strategies are most promising, and whether such risk-framing nudges can improve COVID-19 vaccination uptake. Given that vaccination has become socially and politically charged, affecting hesitancy may no longer be amenable to light-touch framing interventions. Relatedly, it is an open question whether such framing effects will be effective given that individuals have experienced months of side-effect-related media coverage.

To answer these questions, we conducted a randomized controlled trial evaluating the effect of three framing strategies on COVID-19 vaccine intentions among 8988 adults ages 18+ (4502 adults in the United States and 4496 in the United Kingdom). We presented participants with information on a hypothetical future COVID-19 vaccine including information on its side-effect rate (we focused on the risk of serious blood clots and chose a side-effect rate that was comparable to the current vaccines) (*European Medicines Agency (EMA), 2021*; *U.S. Food and Drug, 2021*). We examined three side-effect framing strategies: what is the effect of adding a qualitative risk label next to the numerical risk, what is the effect of adding a comparison group (along with which comparison group is most effective), and for those with comparison groups, what is the effect of framing the comparison in relative rather than absolute terms (absolute comparisons of small risks may be cognitively harder for individuals to process than relative comparisons). Based on a pre-registered and published analysis plan (*Sudharsanan et al., 2021*), we then evaluated the effect of these framing strategies on two outcomes: willingness to take the hypothetical vaccine (Yes/No), and as a measure of the mechanism of our framing effects, individuals' perceived safety of the vaccine (defined on a scale of 1–10). While there is concern that vaccine intentions may not represent actual behavior, two recent papers focused on COVID-19 have shown a strong link between individuals' COVID-19 vaccine intentions and their actual vaccination behavior (*Campos-Mercade et al., 2021*; *Klüver et al., 2021*). For this reason, we believe that the results of our paper can be useful for generating evidence on strategies to slow the COVID-19 pandemic. Understanding how side-effect perceptions influence vaccination is especially important in light of the ongoing booster situation and the low boosting rate even among those that have chosen to receive two doses, as individuals may not choose to be boosted if they perceive the risk of side-effects to outweigh the incremental benefits of additional doses.

## Materials and methods
### Study approvals, registration, and pre-analysis plan
Prior to recruiting any participants, we received ethical approval for the study from the Medical Faculty of Heidelberg University Ethics Committee (#S-443/2021), registered the trial, including the

outcomes and treatments, on the German Clinical Trials Registry (#DRKS00025551), and published a trial protocol with a pre-analysis plan (*Sudharsanan et al., 2021*). All participants provided informed consent through the Prolific platform, including consent to publish.

All the analyses and results we present here are in line with our pre-analysis plan. The study here excludes one supplementary analysis that we registered as part of the protocol and pre-analysis plan. This additional analysis was intended to be based on a comparison of the main study participants (the 8998 individuals who form the current study) to an additional 3000 participants. The reason for the omission of this supplementary analysis is that there was an error in the study text for these additional 3000 participants (the 3000 participants were recruited and successfully completed the experiment; however, we did not conduct our planned supplementary analyses based on the data from these individuals). This error, however, had no impact on the 8998 participants recruited for the main study presented here as these individuals were recruited separately from the main study participants, and as stated in our pre-analysis plan, this excluded analysis was only intended as a supplementary analysis. No harm or adverse events were observed given the online format of the trial.

## Study population, eligibility, inclusion, and exclusion criteria

This was an online-based randomized controlled trial study. To be enrolled in the trial, participants had to meet the following inclusion criteria: be 18 years or older, have current residence in the United States or the United Kingdom, and be able to speak English. Participants who did not meet the inclusion criteria were not eligible to participate and were excluded from the study.

## Participant recruitment

We used Prolific (*Prolific, 2022*), an online-based service for recruiting participants for online-based research studies to recruit the study participants. Prolific participants are paid for their participation (we paid each participant £0.63/$0.88 for the expected 5 min completion time). Within Prolific, we used a stratified quota sampling procedure to match the education distribution of our sample to the general population (stratified by sex and country) based on available data for each country (*Bureau*

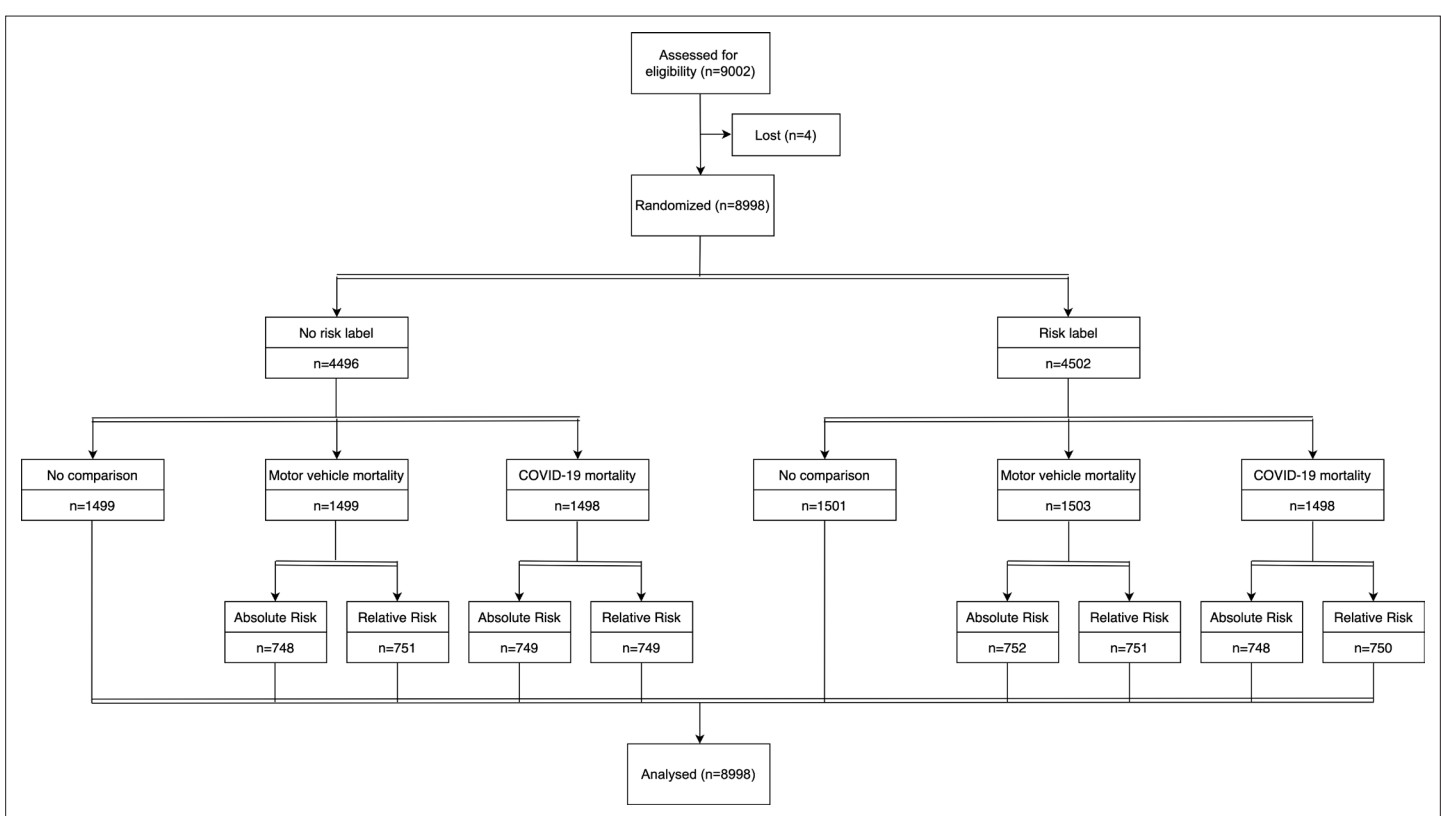

**Figure 1.** CONSORT flow diagram. Of the 9002 enrolled participants, 8998 individuals completed the survey (response rate = 99.96%).

*U.S. Census, 2019*; *OECD, 2021*). We aimed to recruit a total of 9000 total participants, split evenly by country and sex. We chose this sample size to enable us to detect a 5 percentage point difference between the experimental arms with 80% power, a 5% alpha value, and a control proportion of 50%; 99% of our participants were recruited between July 30, 2021, and August 10, 2021. To reach our target quotas by sex and education, the final 1% of participants were recruited between August 11, 2021, and October 4, 2021.

## Description of the experiment

We previously published the protocol for our trial with a description of the experiment (*Sudharsanan et al., 2021*). On the Prolific platform, potential participants were provided information that the aim of the study was to understand their willingness to take COVID-19 vaccines, the risks and benefits of the study, and how they could contact the researcher (and/or the human subjects review board at the Heidelberg University). After consenting on Prolific, participants were redirected to the Gorilla platform, an interactive web-based service for conducting behavioral science experiments. On the Gorilla page, we provided additional information on data protection. For participants that agreed to participate, we first collected basic sociodemographic information and then set up the experiment by telling participants that we are going to show them information on a hypothetical future COVID-19 vaccine and would like to know how willing they would be to take it. At this stage, we emphasized that their answers cannot be linked back to them in any way. Participants were then presented information on the vaccine and its side-effect rate on a single page. The main experimental component was how the vaccine side-effect rate was presented to participants.

## Randomization and blinding

We used a factorial randomization design to assign participants to three main factors (*Figure 1*). Factor 1 was whether there was a qualitative risk label saying 'very low risk' next to the numerical risk or not. Factor 2 was whether the risk was presented with no comparison, a comparison to motor-vehicle mortality, or a comparison to COVID-19 mortality. Factor 3 was only among that received a comparison risk and varied whether the comparison was presented in an absolute or relative way. We randomized participants to each factor independently (stratified by country), such that, for example, which comparison group a participant received did not depend on whether they received a risk label or not. This means that the proportion of participants that received a risk label will be balanced across the treatment groups for the comparison risks. We repeated this independent randomization procedure for whether the comparison risk is presented as an absolute or relative comparison among the two-thirds of the sample that were randomized to receive any comparison risk. Similarly, this means that conditional on receiving any comparison risk (either motor-vehicle or COVID-19 mortality), the proportion of participants that received each type of comparison, and the proportion that received a labeled risk will be balanced across the treatment groups for absolute and relative framings. Note that in practice, the Gorilla algorithm handled the randomization using a stratified randomization approach to prevent chance imbalances in the joint distribution of the factors. Since the recruitment took place on the Prolific platform, it was not possible to identify or link data back to the participants. Participants responded to the survey questions and submitted their responses anonymously through the Gorilla platform. Both the study subjects and the study team members were blinded to the allocation status of the participants.

## Outcomes

Our main outcome was a binary variable for whether participants reported 'yes' to the question of 'Would you take this vaccine if it were made available to you?' Importantly, we asked participants to answer as if they had not been vaccinated even if they already received some form of vaccination. In *Appendix 1—table 2*, we show the main study results using a four-category ordinal response rather than a binary response variable and find no change to our conclusions. Our secondary outcome was participants' perception of how safe the vaccine is, based on a scale of 1 (extremely unsafe) to 10 (extremely safe). This secondary outcome served to determine if the effects we saw on vaccine willingness were at least partly driven by perceptions and judgments of the vaccine's safety.

## Missing data

We had complete data for all 4502 participants from the United States (the extra two participants above our target of 4500 was the result of how Prolific and Gorilla recruit individuals). Four individuals from the UK sample did not complete the experiments and were therefore excluded due to missing data (0.04%). This resulted in a final sample of 8998 individuals (99.96% response rate).

## Statistical analyses

We first assessed if the randomization was successful by comparing means and proportions of the main sociodemographic variables across the experimental arms for each of the randomization factors. Next, we estimated the main effects of the risk label and comparison group strategies using the following logistic regression model:

$$ln\left(\frac{\theta}{1-\theta}\right) = \alpha_0 + (\alpha_1 * CM_i) + (\alpha_2 * MVM_i) + (\alpha_3 * QL_i) + \zeta_i \quad (1)$$

In this model, $\theta$ is the probability of reporting 'Yes' to take the hypothetical vaccine, $CM_i$ is a binary indicator for whether participant $i$ was shown a comparison to COVID-19 mortality, $MVM_i$ is a binary indicator for whether participant $i$ was shown a comparison to motor-vehicle mortality, $QL_i$ is a binary indicator for whether participant $i$ was shown a qualitative risk label, and $\zeta_i$ is a vector of covariates, including age, sex, education, and country. Our main effects of interest are $\alpha_1$, $\alpha_2$, and $\alpha_3$, respectively. We then estimated the effect of a relative compared to absolute comparison framing by estimating the following regression just among those that received either the motor-vehicle or COVID-19 mortality comparison:

$$ln\left(\frac{\theta}{1-\theta}\right) = \beta_0 + (\beta_1 * Rel_i) + \zeta_i \quad (2)$$

Here, $Rel_i$ is a binary indicator for receiving the relative framing and $\beta_1$ is the corresponding effect of this framing on vaccine intentions compared to an absolute comparison.

Based on our pre-registered analyses, we found evidence that adding a qualitative risk label and adding a comparison to motor-vehicle mortality both increased the likelihood that participants reported that they would take the vaccine. In a non-pre-registered analysis, we then assessed whether these two independent effects were additive by estimating the effect of receiving both strategies relative to receiving neither.

To determine whether the main effects varied by country, sex, and age, we re-estimated these two main regressions including interaction terms between the main experimental indicators and the heterogeneity variables (separately for each heterogeneity variable).

**Table 1.** Demographic composition of the study samples.

| | United States (N=4502) | United Kingdom (N=4496) |
|---|---|---|
| **Gender** | | |
| Male | 2,216 (49.2%) | 2,236 (49.7%) |
| Female | 2,205 (49.0%) | 2,216 (49.3%) |
| Other | 81 (1.8%) | 44 (1.0%) |
| **Age** | | |
| 18–30 | 3,090 (68.6%) | 1,859 (41.4%) |
| 30–45 | 1,077 (23.9%) | 1,531 (34.0%) |
| 45–60 | 268 (6.0%) | 776 (17.3%) |
| 60+ | 67 (1.5%) | 330 (7.3%) |
| **Education** | | |
| Less than secondary education | 142 (3.1%) | 498 (11.1%) |
| Completed secondary education | 987 (21.9%) | 1217 (27.1%) |
| Some college | 1786 (39.7%) | 1014 (22.5%) |
| Bachelor's degree | 1061 (23.6%) | 1182 (26.3%) |
| More than bachelor's degree | 526 (11.7) | 585 (13.0%) |
| **Race*** | | |
| White | 3607 (80.1%) | 3813 (84.8%) |
| Black/African | 357 (7.9%) | 139 (3.1%) |
| Asian | 441 (9.8%) | 316 (7.0%) |
| Other | 329 (7.3%) | 190 (4.2%) |
| **Hispanic** | 538 (11.9%) | – |

Notes: *Participants were allowed to choose multiple options if they identified as a member of multiple races. We used the race/ethnic categories used by the US Census Bureau and the UK Office for National Statistics when asking individuals to report their race and/or ethnicity.

**Table 2.** Balance table of baseline participant characteristics between the no risk label and risk label treatment arms.

| | No risk label (*N*=4496) | Risk label (*N*=4502) | p-Value |
|---|---|---|---|
| Female (*n*, %) | 2224 (49.5%) | 2197 (48.8%) | 0.542 |
| Age (mean years, SD) | 31.7 (12.6) | 31.6 (12.7) | 0.623 |
| Education (*n*, %) | | | 0.836 |
| Less than secondary education | 320 (7.1%) | 319 (7.1%) | |
| Completed secondary education | 1097 (24.4%) | 1107 (24.6%) | |
| Some college | 1392 (31.0%) | 1409 (31.3%) | |
| Bachelor's degree | 1143 (25.4%) | 1100 (24.4%) | |
| More than bachelor's degree | 544 (12.1%) | 567 (12.6%) | |

Notes: p-Values for gender and education correspond to a chi-squared test and a two-sided t-test for age.

Lastly, to determine whether these effects were partly mediated by perceptions of vaccine safety, we estimated regressions (1) and (2) using a continuous indicator for reported safety as the main outcome variable.

As robustness and sensitivity analyses, we re-estimated each regression without controls for age, sex, country, and education, using an ordinal logistic regression model with a four-category outcome variable for vaccine intention rather than a binary indicator for 'yes', and using linear probability rather than logistic regression models. To ease the interpretation, we present coefficients as average marginal effects for all logistic regressions (as per our pre-analysis plan, *Sudharsanan et al., 2021*).

## Results

### Demographic composition of the sample

Our study samples contained individuals from a broad range of age, education, and race/ethnic groups (*Table 1*). In the US and UK samples, 68.6% and 41.4% were between ages 18 and 30, 23.9% and 34.0% between 30 and 45, and 7.5% and 24.6% above 45. Similarly, we had representation of those with a high school completion only or lower (25% in the United States and 38.2% in the United Kingdom) and those with a college degree or more (34.3% in the United States and 39.3%

**Table 3.** Balance table of baseline participant characteristics between the no comparison, comparison with motor-vehicle mortality, and comparison with COVID-19 mortality treatment arms.

| | No comparison (*N*=3000) | Comparison with motor-vehicle mortality (*N*=3002) | Comparison with COVID-19 mortality (*N*=2996) | p-Value |
|---|---|---|---|---|
| Female (*n*, %) | 1478 (49.3%) | 1452 (48.4%) | 1491 (49.8%) | 0.547 |
| Age (mean years, SD) | 31.8 (12.8) | 31.4 (12.5) | 31.6 (12.7) | 0.516 |
| Education (*n*, %) | | | | 0.873 |
| Less than secondary education | 210 (7.0%) | 202 (6.7%) | 227 (7.6%) | |
| Completed secondary education | 717 (24.0%) | 744 (24.8%) | 743 (24.8%) | |
| Some college | 931 (31.0%) | 933 (31.1%) | 937 (31.2%) | |
| Bachelor's degree | 769 (25.6%) | 753 (25.1%) | 721 (24.1%) | |
| More than bachelor's degree | 373 (12.4%) | 370 (12.3%) | 368 (12.3%) | |

Notes: p-Values for gender and education correspond to a chi-square test and an ANOVA-test for age.

**Table 4.** Balance table of baseline participant characteristics between the absolute risk and relative risk treatment arms.

| | Absolute risk (*N*=2997) | Relative risk (*N*=3001) | p-Value |
|---|---|---|---|
| Female (*n*, %) | 1473 (49.1%) | 1470 (49.0%) | 0.918 |
| Age (mean years, SD) | 31.4 (12.4) | 31.6 (12.8) | 0.543 |
| Education (*n*, %) | | | 0.897 |
| Less than secondary education | 222 (7.4%) | 207 (6.9%) | |
| Completed secondary education | 737 (24.6%) | 750 (25.0%) | |
| Some college | 934 (31.2%) | 936 (31.2%) | |
| Bachelor's degree | 743 (24.8%) | 731 (24.3%) | |
| More than bachelor's degree | 361 (12.0%) | 377 (12.6%) | |

Notes: p-Values for gender and education correspond to a chi-square test and a two-sided t-test for age.

in the United Kingdom). For race and ethnicity, our samples had representation of individuals who self-identified as Black or African (7.9% in the United States and 3.1% in the United Kingdom), those who identified as Asian (9.8% in the United States and 7.0% in the United Kingdom), and those who identified as Hispanic (11.9% in the United States). As discussed in the following section, among our control group, intentions to vaccinate were 65%, which is closely aligned with the actual vaccination rate in the United States and United Kingdom at the time our data were being collected (*Ritchie et al., 2022*). We also find no evidence of imbalances in sociodemographic characteristics across the experimental groups (*Tables 2–4*).

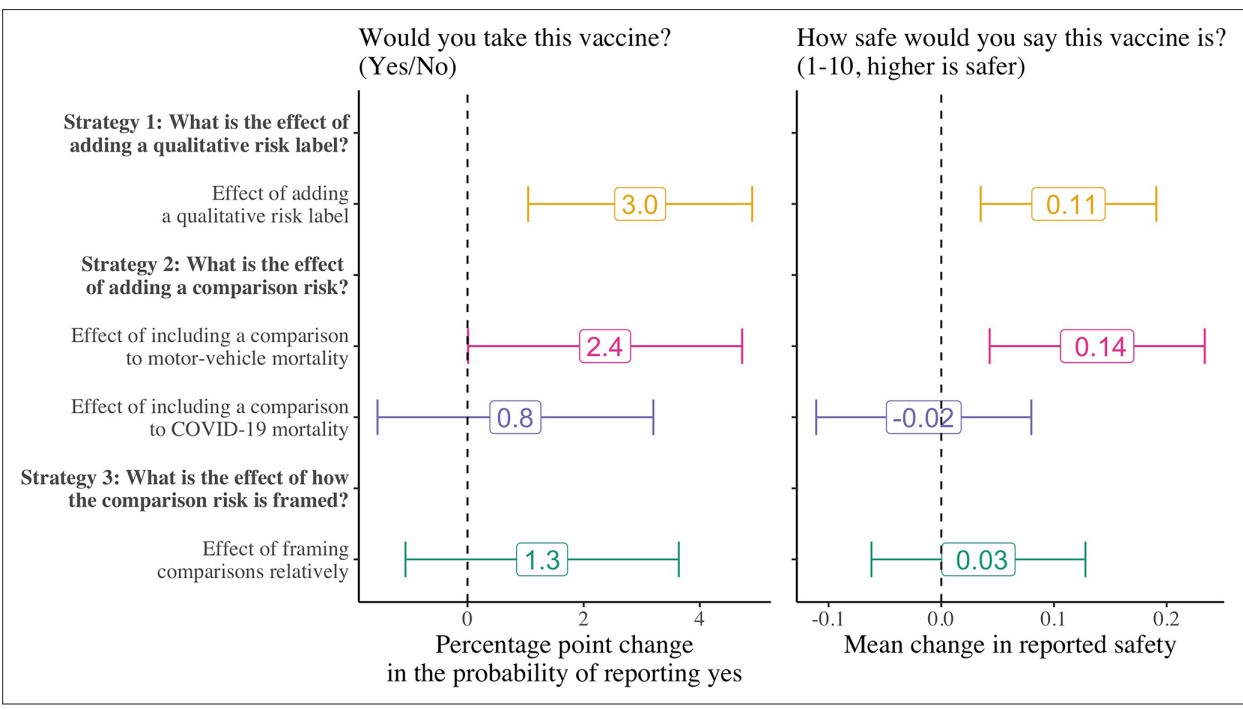

**Figure 2.** Effect of the vaccine side-effect framing strategies on participants' vaccine intentions and perceived safety. *N*=8998 for strategies 1 and 2; strategy 3 is estimated only among those that received a comparison risk (*N*=5998). We estimated the main effects of each treatment strategy on participants' vaccine intentions using logistic regression models and on the perceptions of vaccine safety using OLS regression models. Control means for strategies 1 and 2 are 65% and 67% for strategy 3. Error bars represent 95% confidence intervals.

## Effects of vaccine side-effect framing strategies on COVID-19 vaccine intentions

*Figure 2* presents the results of the three main framing strategies on the proportion of participants that report that they would take the hypothetical vaccine (the results are presented on the percentage point scale). Among all the strategies, we find that adding a simple qualitative risk label ('very low risk') next to the numerical risk increased vaccine intentions by 3.0 percentage points (95% CI: 1.0, 4.9; p=0.003; control group mean: 65%). For our two comparison strategies, we found that adding a comparison to motor-vehicle mortality (effect size: 2.4 percentage points; 95% CI: 0.001, 4.7; p=0.049; control group mean: 65%) had an impact on vaccine intentions but no evidence of an effect of adding a comparison to COVID-19 mortality (effect size: 0.8 percentage points; 95% CI: –1.6, 3.2; p=0.496; control group mean: 65%). This is a surprising result as we expected the comparison to COVID-19 mortality to be more salient both because it is substantially higher than motor-vehicle mortality and because it is the form of mortality directly related to vaccination. In the United States, for example, motor-vehicle mortality in 2020 was 12 per 100,000 population while COVID-19 mortality in the same year was 170 per 100,000 population. Lastly, going against our expectation, we did not find evidence that relative, compared to absolute, framings of comparisons had a large impact on willingness to take the hypothetical future COVID-19 vaccine (effect size: 1.3 percentage points; 95% CI: –1.1, 3.6; p=0.285; control group mean: 67%).

*Figure 2* also presents the effect of the strategies on participants' perceptions of how safe the vaccine is (measured on a scale of 1–10). This secondary outcome serves to investigate whether the effects we observed on vaccine intentions were driven through perceptions of safety (providing more information may affect vaccine intentions, for example, by increasing individuals' trust in the source of information rather than directly affecting their perceptions of vaccine safety). Although the magnitude of the effect sizes is small, we find that the pattern of the framing effects on perceptions of vaccine safety closely mirrors the primary effects on willingness to take the vaccine. This suggests that perceptions of safety are indeed one of the pathways through which these framing effects operate, although our analyses do not exclude the possibility that by providing more information, these framing strategies also affect vaccine intentions by increasing the perceived trustworthiness or reliability of the information.

## Are the two effects substitutes, additive, or synergistic?

An important question is how effective these strategies are when combined. For example, it could be the case that vaccine intentions are only movable by a fixed margin, such that even when both qualitative risk labels and comparison groups are used, the resulting change in vaccine intentions is less than the sum of the independent effects. Conversely, the effects may be synergistic such that combining strategies leads to larger effects than the sum of the independent effects. To assess this

**Table 5.** Independent and combined effects of risk labeling and motor-vehicle mortality comparisons on willingness to take a hypothetical COVID-19 vaccine.

| | Effect size (percentage points) | p-Value |
|---|---|---|
| Effect of risk labeling (ref: no risk label) N=8998 | 3.0 | 0.003 |
| Effect of motor-vehicle comparison (ref: no comparison) N=8998 | 2.4 | 0.049 |
| Effect of both risk labeling and a motor-vehicle comparison (ref: no risk label nor comparison) N=3002 | 6.1 | <0.001 |

*Notes*: Outcome: 'Would you take this vaccine?' (yes = 1, others = 0). All the results are from logistic regression models with the results presented as average marginal effects. As per our pre-analysis plan, all regressions include covariates for age, sex, education, and country. Sample sizes for the relative to absolute comparison and effect of both labeling and a motor-vehicle mortality comparison are smaller since they are only estimated among subset of the total sample. p-Values are from two-tailed t-tests.

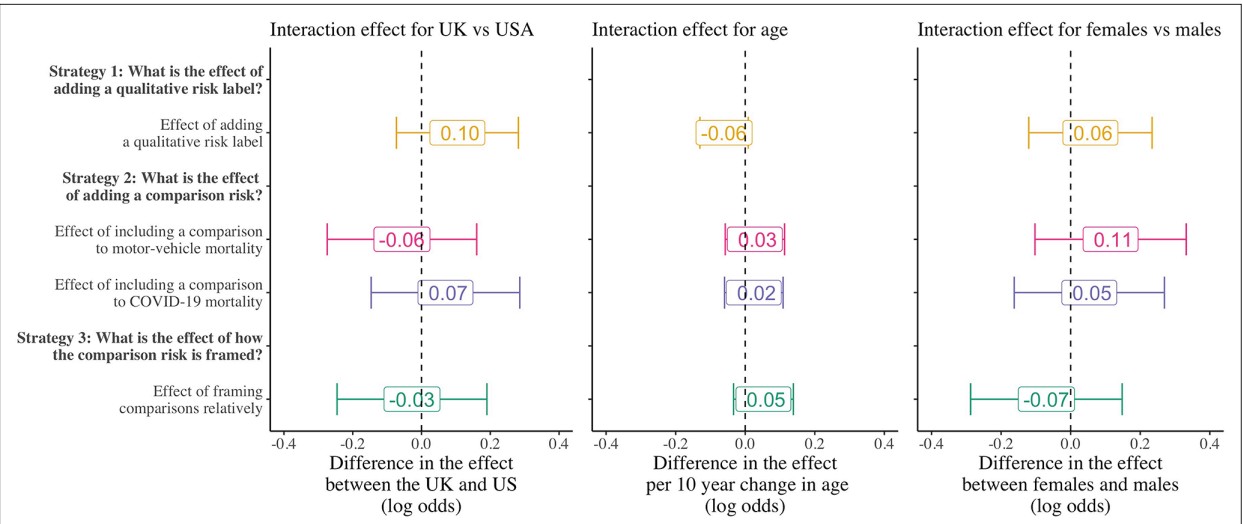

**Figure 3.** Differences in the framing effects by country, age, and sex. Coefficients are the interaction effect of the main heterogeneity characteristic with the indicator for each treatment strategy and are presented as coefficients on the log-odds scale from logistic regression models. Error bars represent 95% confidence intervals.

consideration, in a non-pre-registered analysis, we estimated the effect of adding both a qualitative risk label and comparison to motor-vehicle mortality on willingness to take the hypothetical COVID-19 vaccine (*Table 5*). We find that at a minimum, the effects are independent and additive, with some indication that they may even be synergistic. Compared to those that received neither a quality risk label nor a comparison risk (*N*=1499; control mean: 63%), those that received both strategies were 6.1 percentage points (95% CI: 2.8, 9.5; p<0.001) more likely to state that they would take the vaccine.

## Differences by country, sex, and age group

*Figure 3* examines whether the effects of these framing strategies vary by country, age, and sex. The coefficients in each figure show the interaction effect on the treatment with the main heterogeneity characteristic and are thus interpreted as how much larger is the framing effect for a particular group compared to another on the log-odds scale. There are several reasons to suspect that there may be differences in the magnitude of framing effects across groups. Contexts where vaccines have become highly politicized like the United States may be less responsive to framing effects than in places like the United Kingdom. Older individuals may use different heuristics for assessing risk and thus respond differently to framing effects (although the direction of this is theoretically ambiguous); and on average, men and women may have different risk thresholds for what they deem to be risky or not (*Harris et al., 2006*; *Finucane et al., 2000*; *Alsharawy et al., 2021*), which may affect how the framing effects impact behavior.

We do not find evidence that the framing strategies vary across country, age, or sex. Importantly, due to our sample sizes, we cannot rule out the possibility of potentially small interaction effects. Therefore, our results should be interpreted as not finding evidence for any large differences.

## Robustness

Our results are consistent when we use linear probability rather than logistic regression models (*Appendix 1—table 1*), when we examine the outcome as a four-category ordinal (rather than binary) variable (*Appendix 1—table 2*), and after excluding demographic control variables in the main regression analyses (*Appendix 1—table 1*).

## Discussion

We found that adding a simple descriptive risk label ('very low risk') next to the numerical side-effect increased participants' willingness to take the COVID-19 vaccine by 3.0 percentage points (p=0.003). Providing a comparison to motor-vehicle mortality increased COVID-19 vaccine willingness

by 2.4 percentage points (p=0.049). Importantly, we found that these effects were independent and additive: participants that received both a qualitative risk label and comparison to motor-vehicle mortality were 6.1 percentage points (p<0.001) more likely to report willingness to take a vaccine compared to those who did not receive a label or comparison. This is an important and meaningful effect at the population level, where even small changes in vaccination rates can have large health consequences (*Scobie et al., 2021*). These results are especially reassuring considering the low cost and ease of implementing such framing strategies. Based on the effects on perceptions of vaccine safety, we find support for the hypothesis that these effects work by modifying individuals' judgments about how safe the vaccine is.

One surprising finding was that comparisons to motor-vehicle mortality produced a larger impact on vaccine intentions than comparisons to COVID-19 mortality. One potential explanation for this finding is that at the time of our study, COVID-19 mortality was still a relatively new risk whereas individuals are generally familiar with motor-vehicle mortality. Thus, motor-vehicle mortality may be a more salient heuristic against which individuals evaluate other risks. While speculative, this may suggest that comparisons to commonly understood forms of mortality may produce greater behavioral changes than comparisons to novel or less common risks. A second surprising finding is that we did not find evidence that relative, compared to absolute, framings of comparisons had a larger impact on willingness to take the hypothetical future COVID-19 vaccine. This is in contrast to prior empirical work which generally finds that relative risk framings produce a greater impact on behavior (*Zipkin et al., 2014*). Reconciling these differences will be important for future communication strategies around COVID-19.

## Relationship to the existing and emerging literature

Our results extend and complement a larger literature on using nudges to improve vaccination behavior, especially around COVID-19, and the broader literature on communicating risks to patients (*Trevena et al., 2013*). Campos-Mercade et al. find that among Swedish adults, paying individuals 24 USD increased vaccination rates by 4.2 percentage points but found no evidence of impact from nudges that tried to influence behavior through social network effects and health information (*Campos-Mercade et al., 2021*). Klüver et al. also study financial incentives and find a weaker effect among German adults when incentives were 25 euros, but evidence of moderate effects (between 2.2 and 3 percentage points) on vaccine intentions from larger payments (50 euros), providing freedoms to vaccinated individuals, and from allowing vaccination at local doctors (*Klüver et al., 2021*). Dai et al. conduct a trial among adults enrolled in the UCLA health system and find that text-messaged-based reminders with content based around different behavioral theories increased vaccination rates by 3.57 percentage points (*Dai et al., 2021*). Santos et al. study email-delivered nudges to patients in a large Pennsylvania health system and find that emails that emphasize social norms and similar to one of our main framings, those that reframe COVID-19 risks, increased appointments for vaccination by around 3.5 percentage points (*Santos et al., 2021*). Similarly, a multicountry randomized experiment by Moehring et al. finds that providing individuals information about the share of their network that is accepting of vaccines influences individuals' own vaccination intentions (*Moehring et al., 2000*). While not focused on COVID-19 vaccination, two recent studies by Milkman et al. find that text-message-based nudges leveraging loss-aversion and social pressure can increase flu vaccination by up to 5 percentage points (*Milkman et al., 2021b*; *Milkman et al., 2021a*). Taken together, our results contribute to research on behavioral science strategies for addressing COVID-19 vaccine hesitancy by revealing changes to side-effect framing as another additional nudge-based pathway for impact.

## Limitations

Using self-reported intention to take the vaccine is potentially an important limitation if vaccine intentions do not translate into actual vaccine behavior. Two recent papers focused on COVID-19 that have measured both intentions and actual vaccination decisions have shown a strong link between individuals' COVID-19 vaccine intentions and behavior (*Campos-Mercade et al., 2021*; *Klüver et al., 2021*), providing confidence that the results here can provide important information regarding actual vaccination behavior. However, there is always a possibility that the effects observed here would be smaller or not present when measuring actual behavior. Relatedly, our experiment deals with a hypothetical COVID-19 vaccine rather than one of the existing vaccines currently in use. While we made

this decision to guard against participant bias toward existing vaccines, the framing strategies we present here may not apply to current vaccines if individuals' existing views hinder behavior change. We were not able to distinguish between those that had already been vaccinated – either with one or two doses – and those that were yet to take any vaccine. It could be that the effects here are being driven by those that have already received some vaccines and that our framing effects may not affect the behavior of the unvaccinated. While this is an important consideration, we believe focusing on all adults is particularly important due to the ongoing situation with booster vaccines and proposals for continuing vaccine doses. Specifically, as the number of recommended doses increases, those that have received, for example just two doses, will need to be motivated to take additional doses. The low rates of boosting among those who already received two doses in the United States (41.7%, *Boas et al., 2020*) and the less than ideal rate in the United Kingdom (77.1%, *UK Health Security Agency, 2022*) reveal that just because individuals took two doses, they are not guaranteed to make the decision to take subsequent recommended doses. This is especially the case if they perceive the incremental benefits of additional doses to be outweighed by potential side-effects. For this reason, we believe our focus on all adults, rather than on just those that have not received any vaccine is justified as strategies to encourage COVID-19 vaccination and help individuals better weigh benefits against perceived risks must also focus on those that have received one, two, and potentially in the future, three or more doses.

Our use of an online-recruited sample has the potential to generate selection bias in two important ways. First, the data were collected through online sampling and thus may not be representative of the US and UK adult populations. This is an important potential limitation. We tried to address this potential issue by recruiting individuals from several age groups, race/ethnic groups, and by matching the education distribution of the general population. This especially addresses the concern that online samples tend to be more educated relative to the overall populations (*Boas et al., 2020*; *Paolacci et al., 2010*). However, relative to the general US and UK adult populations, our sample had a slight over-representation of younger adults, Asian Americans, and White Americans and a slight under-representation of older adults, Black Americans, and Hispanic Americans. Since our goal was not to generate a representative descriptive estimate but rather to estimate the effect of different framings, a lack of representativity would only change our findings if those included in the study have a substantially different response to the framings than those that were not included. Our findings suggest that large differences in the main framing effects across social groups are unlikely but this is still an important consideration, especially if there are smaller differences between those that were and were not included in the study that may still be meaningful at a population level. Second, Prolific recruits participants by launching a study on their platform and waiting for registered Prolific users to self-select into participating. Thus, our sample may over-represent individuals with interests in our study topic. Prolific attempts to reduce this bias by randomly selecting a subset of eligible participants and sending them invitations to participate every 48 hr until the sample size is reached. Regardless, it is unlikely that this type of selection bias will have affected our results as randomization uniformly distributes topic-specific selection bias across the trial arms.

## Conclusions

Our results reveal that despite increasingly strong vaccination hesitancy and exposure to large amounts of vaccine information, low-cost side-effect framing strategies can meaningfully affect vaccination intentions at a population level. Given that vaccination for COVID-19 will likely remain an important priority for the foreseeable future, these insights can be valuable for increasing the uptake of future vaccination efforts.

## Additional information

### Competing interests

Till Bärnighausen: TB received consultancy fees for KfW on the OSCAR initiative in Vietnam. TB also participated in the following: NIH-funded study "Healthy Options", Chair of the Data Safety and Monitoring Board (DSMB), German National Committee on the "Future of Public Health Research and Education", Chair of the scientific advisory board to the EDCTP Evaluation, Member of the UNAIDS

Evaluation Expert Advisory Committee, National Institutes of Health Study Section Member on Population and Public Health Approaches to HIV/AIDS (PPAH), US National Academies of Sciences, Engineering, and Medicine's Committee for the "Evaluation of Human Resources for Health in the Republic of Rwanda under the President's Emergency Plan for AIDS Relief (PEPFAR)", University of Pennsylvania (UPenn) Population Aging Research Center (PARC) External Advisory Board Member, Co-chair of the Global Health Hub Germany (which was initiated by the German Ministry of Health). The author has no other competing interests to declare. The other authors declare that no competing interests exist.

### Funding
No external funding was received for this work.

### Author contributions
Nikkil Sudharsanan, Conceptualization, Formal analysis, Supervision, Investigation, Visualization, Methodology, Writing – original draft, Writing – review and editing; Caterina Favaretti, Data curation, Software, Formal analysis, Project administration, Writing – review and editing; Violetta Hachaturyan, Data curation, Software, Formal analysis, Visualization, Project administration, Writing – review and editing; Till Bärnighausen, Conceptualization, Resources, Supervision, Funding acquisition, Investigation, Methodology, Writing – review and editing; Alain Vandormael, Conceptualization, Formal analysis, Supervision, Validation, Investigation, Methodology, Writing – review and editing

### Author ORCIDs
Nikkil Sudharsanan ⓘ http://orcid.org/0000-0003-1710-4634
Caterina Favaretti ⓘ http://orcid.org/0000-0001-7448-3736
Alain Vandormael ⓘ http://orcid.org/0000-0002-5742-0511

### Ethics
Registration German Clinical Trials Registry (#DRKS00025551).
Prior to recruiting any participants, we received ethical approval for the study from the Medical Faculty of Heidelberg University Ethics Committee (#S-443/2021).

### Decision letter and Author response
Decision letter https://doi.org/10.7554/eLife.78765.sa1
Author response https://doi.org/10.7554/eLife.78765.sa2

---

# Additional files

### Supplementary files
• MDAR checklist

### Data availability
The datasets generated during and/or analyzed during the current study and the analysis codes used to produce the figures and tables are available in the Open Science Framework repository, https://doi.org/10.17605/OSF.IO/HQNKR.

The following dataset was generated:

| Author(s) | Year | Dataset title | Dataset URL | Database and Identifier |
|---|---|---|---|---|
| Sudharsanan N | 2021 | Effects of Side-Effect Risk Framing Strategies on COVID-19 Vaccine Intentions: A Randomized Controlled Trial | https://doi.org/10.17605/OSF.IO/HQNKR | Open Science Framework, 10.17605/OSF.IO/HQNKR |

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

# Appendix 1

A) The United States Food and Drug Administration (FDA) has just approved a new COVID-19 vaccine. Based on clinical trials, this vaccine is 95% effective against infection from SARS-CoV-2 (the virus that causes COVID-19), including against the Delta variant.

*With regards to side effects, 1 out of 100,000 vaccinated individuals may develop serious blood clots.*

Would you take this vaccine if it were made available to you (even if you have already been vaccinated, please answer as if you were not yet vaccinated)?

◯ Yes
◯ Unsure - leaning towards yes
◯ Unsure - leaning towards no
◯ No

How safe do you think this vaccine is, on a scale of 1 (=Extremely unsafe) to 10 (=Extremely safe)?

**Extremely unsafe** | 1 | 2 | 3 | 4 | 5 | 6 | 7 | 8 | 9 | 10 | **Extremely safe**

B) The United States Food and Drug Administration (FDA) has just approved a new COVID-19 vaccine. Based on clinical trials, this vaccine is 95% effective against infection from SARS-CoV-2 (the virus that causes COVID-19), including against the Delta variant.

*With regards to side effects, 1 out of 100,000 vaccinated individuals may develop serious blood clots (very low risk). As a reference, 12 out of every 100,000 Americans died in a motor vehicle accident based on data from the past year.*

Would you take this vaccine if it were made available to you (even if you have already been vaccinated, please answer as if you were not yet vaccinated)?

◯ Yes
◯ Unsure - leaning towards yes
◯ Unsure - leaning towards no
◯ No

How safe do you think this vaccine is, on a scale of 1 (=Extremely unsafe) to 10 (=Extremely safe)?

**Extremely unsafe** | 1 | 2 | 3 | 4 | 5 | 6 | 7 | 8 | 9 | 10 | **Extremely safe**

**Appendix 1—figure 1.** Screenshots of the online, Gorilla-based, experiment content for 2 of the 10 possible experimental combinations (A) no risk label, no control group; (B) risk label, comparison to motor-vehicle mortality, absolute comparison.

**Appendix 1—table 1.** Robustness of main paper results to alternative regression types and the inclusion/exclusion of covariates.

| | Logistic regression models with results presented as average marginal effects | | Linear probability models | |
|---|---|---|---|---|
| | **With covariates (main paper results)** | **Without covariates** | **With covariates** | **Without covariates** |
| Effect of labeled risk compared to unlabeled risk | 3.0 pp (p=0.003) | 3.0 pp (p=0.003) | 3.0 pp (p=0.003) | 3.0 pp (p=0.003) |
| Effect of comparison to motor-vehicle mortality compared to no comparison | 2.4 pp (p=0.049) | 2.4 pp (p=0.051) | 2.4 pp (p=0.0498) | 2.4 pp (p=0.052) |
| Effect of comparison to COVID-19 mortality compared to no comparison | 0.8 pp (p=0.496) | 0.7 pp (p=0.568) | 0.8 pp (p=0.497) | 0.7 pp (p=0.571) |
| Effect of relative comparison compared to absolute comparison | 1.3 pp (p=0.285) | 1.3 pp (p=0.277) | 1.3 pp (p=0.286) | 1.3 pp (p=0.277) |
| Effect of both risk labeling and a motor-vehicle comparison compared to no labeling or comparison | 6.1 pp (p<0.001) | 6.1 pp (p<0.001) | 6.1 pp (p<0.001) | 6.1 pp (p<0.001) |

*Notes*: Outcome: 'Would you take this vaccine?' (yes = 1, others = 0). As per our pre-analysis plan, covariates include age, sex, education, and country. Sample sizes for the relative to absolute comparison (*N*=5998) and effect of both labeling and a motor-vehicle mortality comparison (*N*=3002) are smaller since they are only estimated among subset of the total sample.

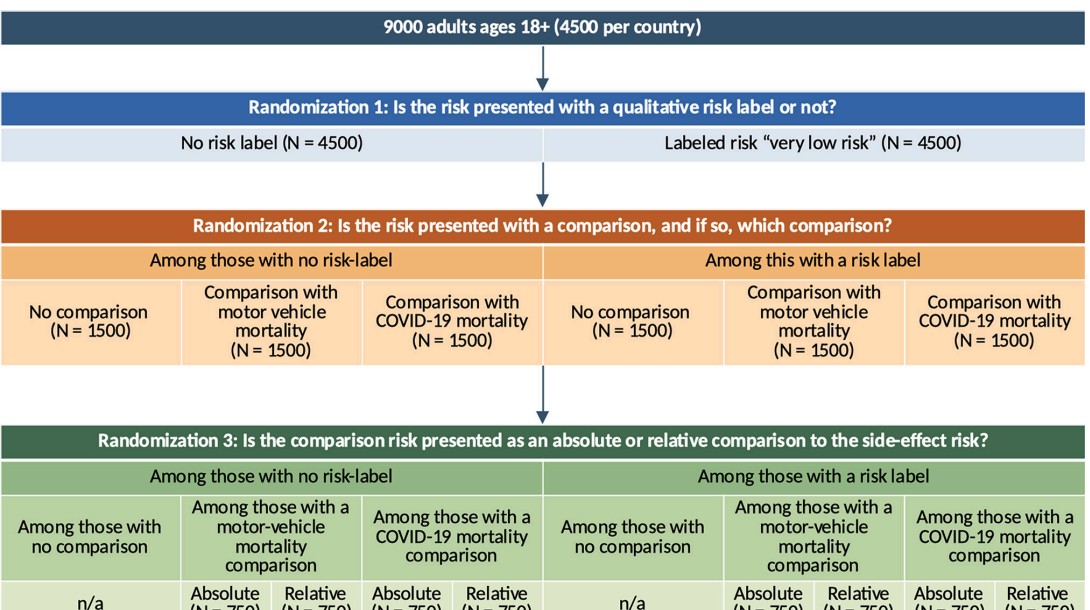

**Appendix 1—figure 2.** Detailed randomization diagram across the main study factors. We have shown the randomization here with sample sizes referring to both countries combined; in practice, we conducted this procedure stratified by country such that each cell in the diagram has exactly 50% US and 50% UK observation.

**Appendix 1—table 2.** Main paper results estimated using ordinal logistic regression models with a four-category outcome specification rather than logistic regression models with a binary outcome.

|  | Odds ratio (p-value) |
| --- | --- |
| Effect of labeled risk compared to unlabeled risk | 1.15 (p=0.002) |
| Effect of comparison to motor-vehicle mortality compared to no comparison | 1.11 (p=0.050) |
| Effect of comparison to COVID-19 mortality compared to no comparison | 1.03 (p=0.541) |
| Effect of relative comparison compared to absolute comparison | 1.05 (p=0.327) |
| Effect of both risk labeling and a motor-vehicle comparison compared to no labeling or comparison | 1.31 (p<0.001) |

*Note.* Outcome: 'Would you take this vaccine?' (No, Unsure – leaning towards no, Unsure – leaning towards yes, Yes). As per the main paper results and pre-analysis plan, all regressions include controls for age, sex, country, and education. Sample sizes for the relative to absolute comparison (*N*=5998) and effect of both labeling and a motor-vehicle mortality comparison (*N*=3002) are smaller since they are only estimated among subset of the total sample.

**Arm 1**

The United States Food and Drug Administration (FDA) has just approved a new COVID-19 vaccine. Based on clinical trials, this vaccine is 95% effective against infection from SARS-CoV-2 (the virus that causes COVID-19), including against the Delta variant.

*With regards to side effects, 1 out of 100,000 vaccinated individuals may develop serious blood clots.*

**Arm 2**

The United States Food and Drug Administration (FDA) has just approved a new COVID-19 vaccine. Based on clinical trials, this vaccine is 95% effective against infection from SARS-CoV-2 (the virus that causes COVID-19), including against the Delta variant.

*With regards to side effects, 1 out of 100,000 vaccinated individuals may develop serious blood clots. As a reference, 12 out of every 100,000 Americans died in a motor vehicle accident based on data from the past year.*

**Arm 3**

The United States Food and Drug Administration (FDA) has just approved a new COVID-19 vaccine. Based on clinical trials, this vaccine is 95% effective against infection from SARS-CoV-2 (the virus that causes COVID-19), including against the Delta variant.

*With regards to side effects, 1 out of 100,000 vaccinated individuals may develop serious blood clots. As a reference, this is 1/12th of the risk of dying in a motor vehicle accident based on data from the past year.*

**Arm 4**

The United States Food and Drug Administration (FDA) has just approved a new COVID-19 vaccine. Based on clinical trials, this vaccine is 95% effective against infection from SARS-CoV-2 (the virus that causes COVID-19), including against the Delta variant.

*With regards to side effects, 1 out of 100,000 vaccinated individuals may develop serious blood clots. As a reference, 170 out of every 100,000 unvaccinated Americans died of COVID-19 based on data from the past year.*

**Arm 5**

The United States Food and Drug Administration (FDA) has just approved a new COVID-19 vaccine. Based on clinical trials, this vaccine is 95% effective against infection from SARS-CoV-2 (the virus that causes COVID-19), including against the Delta variant.

*With regards to side effects, 1 out of 100,000 vaccinated individuals may develop serious blood clots. As a reference, this is 1/170th of the risk of COVID-19 mortality among unvaccinated Americans based on data from the past year.*

**Arm 6**

The United States Food and Drug Administration (FDA) has just approved a new COVID-19 vaccine. Based on clinical trials, this vaccine is 95% effective against infection from SARS-CoV-2 (the virus that causes COVID-19), including against the Delta variant.

*With regards to side effects, 1 out of 100,000 vaccinated individuals may develop serious blood clots (very low risk).*

**Arm 7**

The United States Food and Drug Administration (FDA) has just approved a new COVID-19 vaccine. Based on clinical trials, this vaccine is 95% effective against infection from SARS-CoV-2 (the virus that causes COVID-19), including against the Delta variant.

*With regards to side effects, 1 out of 100,000 vaccinated individuals may develop serious blood clots (very low risk). As a reference, 12 out of every 100,000 Americans died in a motor vehicle accident based on data from the past year.*

**Arm 8**

The United States Food and Drug Administration (FDA) has just approved a new COVID-19 vaccine. Based on clinical trials, this vaccine is 95% effective against infection from SARS-CoV-2 (the virus that causes COVID-19), including against the Delta variant.

*With regards to side effects, 1 out of 100,000 vaccinated individuals may develop serious blood clots (very low risk). As a reference, this is 1/12th of the risk of dying in a motor vehicle accident based on data from the past year.*

**Arm 9**

The United States Food and Drug Administration (FDA) has just approved a new COVID-19 vaccine. Based on clinical trials, this vaccine is 95% effective against infection from SARS-CoV-2 (the virus that causes COVID-19), including against the Delta variant.

*With regards to side effects, 1 out of 100,000 vaccinated individuals may develop serious blood clots (very low risk). As a reference, 170 out of every 100,000 unvaccinated Americans died of COVID-19 based on data from the past year.*

**Arm 10**

The United States Food and Drug Administration (FDA) has just approved a new COVID-19 vaccine. Based on clinical trials, this vaccine is 95% effective against infection from SARS-CoV-2 (the virus that causes COVID-19), including against the Delta variant.

*With regards to side effects, 1 out of 100,000 vaccinated individuals may develop serious blood clots (very low risk). As a reference, this is 1/170th of the risk of COVID-19 mortality among unvaccinated Americans based on data from the past year.*

**Appendix 1—figure 3.** Screenshots of each experimental arm for US participants. Note that we have labeled the arms for the figure, but participants were not shown this label. Reference mortality information for motor-vehicle and COVID-19 fatalities were taken from the Centers for Disease Control and Prevention (*Centers for Disease Control and Prevention, 2021b*; *Centers for Disease Control and Prevention, 2020*).

**Arm 1**

The Medicines and Healthcare products Regulatory Agency (MHRA) has just approved a new COVID-19 vaccine. Based on clinical trials, this vaccine is 95% effective against infection from SARS-CoV-2 (the virus that causes COVID-19), including against the Delta variant.

*With regards to side effects, 1 out of 100,000 vaccinated individuals may develop serious blood clots.*

**Arm 2**

The Medicines and Healthcare products Regulatory Agency (MHRA) has just approved a new COVID-19 vaccine. Based on clinical trials, this vaccine is 95% effective against infection from SARS-CoV-2 (the virus that causes COVID-19), including against the Delta variant.

*With regards to side effects, 1 out of 100,000 vaccinated individuals may develop serious blood clots. As a reference, 2.6 out of every 100,000 individuals in the UK died in a motor vehicle accident based on data from the past year.*

**Arm 3**

The Medicines and Healthcare products Regulatory Agency (MHRA) has just approved a new COVID-19 vaccine. Based on clinical trials, this vaccine is 95% effective against infection from SARS-CoV-2 (the virus that causes COVID-19), including against the Delta variant.

*With regards to side effects, 1 out of 100,000 vaccinated individuals may develop serious blood clots. As a reference, this is almost 1/4th of the risk of dying in a motor vehicle accident based on data from the past year.*

**Arm 4**

The Medicines and Healthcare products Regulatory Agency (MHRA) has just approved a new COVID-19 vaccine. Based on clinical trials, this vaccine is 95% effective against infection from SARS-CoV-2 (the virus that causes COVID-19), including against the Delta variant.

*With regards to side effects, 1 out of 100,000 vaccinated individuals may develop serious blood clots. As a reference, 108 out of every 100,000 unvaccinated individuals in the UK died of COVID-19 based on data from the past year.*

**Arm 5**

The Medicines and Healthcare products Regulatory Agency (MHRA) has just approved a new COVID-19 vaccine. Based on clinical trials, this vaccine is 95% effective against infection from SARS-CoV-2 (the virus that causes COVID-19), including against the Delta variant.

*With regards to side effects, 1 out of 100,000 vaccinated individuals may develop serious blood clots. As a reference, this is 1/108th of the risk of COVID-19 mortality among unvaccinated individuals in the UK based on data from the past year.*

**Arm 6**

The Medicines and Healthcare products Regulatory Agency (MHRA) has just approved a new COVID-19 vaccine. Based on clinical trials, this vaccine is 95% effective against infection from SARS-CoV-2 (the virus that causes COVID-19), including against the Delta variant.

*With regards to side effects, 1 out of 100,000 vaccinated individuals may develop serious blood clots (very low risk).*

**Arm 7**

The Medicines and Healthcare products Regulatory Agency (MHRA) has just approved a new COVID-19 vaccine. Based on clinical trials, this vaccine is 95% effective against infection from SARS-CoV-2 (the virus that causes COVID-19), including against the Delta variant.

*With regards to side effects, 1 out of 100,000 vaccinated individuals may develop serious blood clots (very low risk). As a reference, 2.6 out of every 100,000 individuals in the UK died in a motor vehicle accident based on data from the past year.*

**Arm 8**

The Medicines and Healthcare products Regulatory Agency (MHRA) has just approved a new COVID-19 vaccine. Based on clinical trials, this vaccine is 95% effective against infection from SARS-CoV-2 (the virus that causes COVID-19), including against the Delta variant.

*With regards to side effects, 1 out of 100,000 vaccinated individuals may develop serious blood clots (very low risk). As a reference, this is nearly 1/4th of the risk of dying in a motor vehicle accident based on data from the past year.*

**Arm 9**

The Medicines and Healthcare products Regulatory Agency (MHRA) has just approved a new COVID-19 vaccine. Based on clinical trials, this vaccine is 95% effective against infection from SARS-CoV-2 (the virus that causes COVID-19), including against the Delta variant.

*With regards to side effects, 1 out of 100,000 vaccinated individuals may develop serious blood clots (very low risk). As a reference, 108 out of every 100,000 unvaccinated individuals in the UK died of COVID-19 based on data from the past year.*

**Arm 10**

The Medicines and Healthcare products Regulatory Agency (MHRA) has just approved a new COVID-19 vaccine. Based on clinical trials, this vaccine is 95% effective against infection from SARS-CoV-2 (the virus that causes COVID-19), including against the Delta variant.

*With regards to side effects, 1 out of 100,000 vaccinated individuals may develop serious blood clots (very low risk). As a reference, this is 1/108th of the risk of COVID-19 mortality among unvaccinated individuals in the UK based on data from the past year.*

**Appendix 1—figure 4.** Screenshots of each experimental arm for UK participants. Note that we have labeled the arms for the figure, but participants were not shown this label. Reference mortality information for motor-vehicle fatalities were taken from the Department of Transportation (***Department for Transport, 2020***) and COVID-19 from the Public Health England (***Public Health England, 2022***).

