## [Editor Report]

This timely online randomized clinical trial is based on 8,998 participants from the U.S. and the U.K. to examine the association between risk-framing nudges and the willingness to get a Covid vaccine. This manuscript would be of interest to behavioral scientists, particularly behavioral economists. Findings from this work indicate that framing matters and can substantially increase the willingness to get vaccinated.

---

## [Decision Letter]

**Decision letter after peer review:**

Thank you for submitting your article "Effects of Side-Effect Risk Framing Strategies on COVID-19 Vaccine Intentions in the United States and the United Kingdom: A Randomized Controlled Trial" for consideration by *eLife*. Your article has been reviewed by 3 peer reviewers, one of is a member of our Board of Reviewing Editors, and the evaluation has been overseen by Betty Diamond as the Senior Editor. The reviewers have opted to remain anonymous.

Essential revisions:

1) Addressing potential selection bias among participants. Please add a table to describe the characteristics of participants in each arm of the trial.

2) Please explain how the authors handle missing data? Is there any information regarding the 3,000 participants who did not participate?

3) Please add more information to the methods section regarding the regression analysis.

*Reviewer #1 (Recommendations for the authors):*

1. Please present the schematic flow diagram of the sample. Please provide more information about the followings:

2. What was the response rate?

3. How did you handle missing data?

4. Any information about the 3,000 people who did not participate?

5. How did the authors minimize the selection bias in their sampling strategy?

6. Did the author collect any information regarding the proximate geographic location of the participants (i.e., state or region in the country of residents)?

Methods and Statistical Analysis:

1. Please provide more information regarding the regression analysis used. This information is completely missing from the methods section.

Results:

I would like to see a descriptive table among participants in different arms of the trial. In Table 1, participants were clustered by their country of residents. Please provide a table showing the characteristics of people who receive different framing messages.

*Reviewer #2 (Recommendations for the authors):*

1. (Lines 4-6, Page 6, last paragraph of Introduction) Although available in the Results section, an early description of the types of two outcomes (e.g., binary or categorical) and what values the outcomes can take (e.g., 0/1 or counts) in the Introduction section would be helpful.

2. In Figure 1, authors are suggested to (1) add "(Yes/No)" after the header "Would you take this vaccine?" (left panel) or use the description in Table 2, and (2) move "(1-10, higher is safer)" to the end of the header "How safe would you say this vaccine is?" (right panel). The current location of "(1-10, higher is safer)", which is right below "Mean change in reported safety" may confuse the readers.

3. In the captions of Figures 1 and 2 and Table 2, it would be helpful to briefly mention the regression models used.

4. (Lines 1-2, Page 20) Authors should use caution when using the last statement in the Limitations section. As acknowledged early, no meaningful heterogeneity could be largely due to the limited sample size of the study.

*Reviewer #3 (Recommendations for the authors):*

For future work, I would like to see a cluster analysis of participants stratified by age. To see how the framing strategies affect middle-aged adults ages 45-64 or older adults ages 65 and above.

---

## [Author Response]

Essential revisions:1) Addressing potential selection bias among participants. Please add a table to describe the characteristics of participants in each arm of the trial.

We have revised our manuscript to include tables that describe the participants in each arm of the trial along with a p-value testing whether there are differences in characteristics across arms (Tables 2-4; these tables were previously in the Supplemental Material and have now been moved to the main manuscript).

We have also expanded our Limitations subsection to include a more detailed discussion of the possible biases that may arise from a sampling strategy such as ours that relies on an online sample recruited from the Prolific platform:

“Our use of an online-recruited sample has the potential to generate selection bias in two important ways. First, the data were collected through online sampling and thus may not be representative of the US and UK adult populations. We tried to address this potential issue by recruiting individuals from several age groups, race/ethnic groups, and by matching the education distribution of the general population. This especially addresses the concern that online samples tend to be more educated relative to the overall populations (Boas et al., 2020; Paolacci et al., 2010). However, relative to the general US and UK adult populations, our sample had a slight over-representation of younger adults, Asian Americans, and White Americans and a slight under-representation of older adults, Black Americans, and Hispanic Americans. Since our goal was not to generate a representative descriptive estimate but rather to estimate the effect of different framings, a lack of representativity would only change our findings if those included in the study have a substantially different response to the framings than those that were not included. We believe this is unlikely given our own findings of no meaningful heterogeneity in the main study effect by age, sex, or education. Second, Prolific recruits participants by launching a study on their platform and waiting for registered Prolific users to self-select into participating. Thus our sample may over-represent individuals with interests in our study topic. Prolific attempts to reduce this bias by randomly selecting a subset of eligible participants and sending them invitations to participate every 48 hours until the sample size is reached. Regardless, it is unlikely that this type of selection bias will have affected our results as randomization uniformly distributes topic-specific selection bias across the trial arms.”

2) Please explain how the authors handle missing data? Is there any information regarding the 3,000 participants who did not participate?

We apologize for the confusion around the additional 3000 participants described in the pre-analysis plan but not included in the study. These individuals did not refuse participation, rather they were recruited in addition to the main study participants and randomized into an additional study arm that we had initially planned as a secondary analysis (all the individuals randomized to this additional arm participated and completed the experiment). However, due to a mistake in the experimental text for this specific arm, we were not able to conduct this planned supplementary analysis. Thus, excluding these individuals did not introduce any bias into our main results as they were recruited separately for an additional analysis that we did not end up pursuing or completing. The reason we described this issue in the manuscript is to help readers reconcile the difference between our pre-analysis plan and the analysis presented here. We have revised the text to provide a clearer explanation:

“The study here excludes one supplementary analysis that we registered as part of the protocol and pre-analysis plan. This additional analysis was intended to be based on a comparison of the main study participants (the 8998 individuals who form the current study) to an additional 3000 participants. The reason for the omission of this supplementary analysis is that there was an error in the study text for these additional 3000 participants (the 3000 participants were recruited and successfully completed the experiment; however, we did not conduct our planned supplementary analyses based on the data from these individuals). This error, however, had no impact on the 8998 participants recruited for the main study presented here as these individuals were recruited separately from the main study participants, and as stated in our pre-analysis plan, this excluded analysis was only intended as a supplementary analysis.”

For missingness, of the 9002 enrolled participants, just 4 did not complete the survey. We excluded these observations from the analysis and believe that it is unlikely that dropping such a low percentage of missing values (0.04%) led to a bias in our results. We describe this in the Missing Data subsection of the Materials and methods section, copied here:

“Missing Data

We had complete data for all 4502 participants from the USA (the extra two participants above our target of 4500 was the result of how Prolific and Gorilla recruit individuals). 4 individuals from the UK sample did not complete the experiments and were therefore excluded due to missing data (0.04%). This resulted in a final sample of 8998 individuals (99.96% response rate).”

3) Please add more information to the methods section regarding the regression analysis.

The Statistical Analyses subsection of the Materials and methods section describes the regression analysis in detail.

Reviewer #1 (Recommendations for the authors):1. Please present the schematic flow diagram of the sample.

We present the flow diagram in Figure 3 of the manuscript.

2. What was the response rate?

Of the 9002 enrolled participants, 8998 individuals completed the survey. The response rate was 99.9%. We have added this information in the caption of Figure 3 and in the text in the Missing Data subsection of the Materials and methods section.

3. How did you handle missing data?

Please see our response to Esssential Revision #2

4. Any information about the 3,000 people who did not participate?

Please see our response to Esssential Revision #2.

5. How did the authors minimize the selection bias in their sampling strategy?

We discuss in detail the possible selection biases generated from the Prolific platform, as well as the strategies to attenuate them, in our response to Essential Revisions #1.

6. Did the author collect any information regarding the proximate geographic location of the participants (i.e., state or region in the country of residents)?

We appreciate the reviewer’s comment and agree that it would be interesting to understand whether the framing effects vary by state or region of residence. Unfortunately we did not collect this information since we did not have a large enough sample size to discuss regional differences.

Methods and Statistical Analysis:1. Please provide more information regarding the regression analysis used. This information is completely missing from the methods section.

The Statistical Analyses subsection of the Materials and methods section describes the regression analysis in detail. Please see our response to Essential Revision #3.

Results:I would like to see a descriptive table among participants in different arms of the trial. In Table 1, participants were clustered by their country of residents. Please provide a table showing the characteristics of people who receive different framing messages.

Please see our response to Essential Revision #1

Reviewer #2 (Recommendations for the authors):1. (Lines 4-6, Page 6, last paragraph of Introduction) Although available in the Results section, an early description of the types of two outcomes (e.g., binary or categorical) and what values the outcomes can take (e.g., 0/1 or counts) in the Introduction section would be helpful.

We thank the reviewer for this helpful suggestion and have revised the sentence accordingly:

“…Based on a pre-registered and published analysis plan (32), we then evaluated the effect of these framing strategies on two outcomes: willingness to take the hypothetical vaccine (Yes/No), and as a measure of the mechanism of our framing effects, individuals’ perceived safety of the vaccine (defined on a scale of 1-10).”

2. In Figure 1, authors are suggested to (1) add "(Yes/No)" after the header "Would you take this vaccine?" (left panel) or use the description in Table 2, and (2) move "(1-10, higher is safer)" to the end of the header "How safe would you say this vaccine is?" (right panel). The current location of "(1-10, higher is safer)", which is right below "Mean change in reported safety" may confuse the readers.

Thank you for this suggestion, we have revised the figure accordingly.

3. In the captions of Figures 1 and 2 and Table 2, it would be helpful to briefly mention the regression models used.

Following the reviewer’s comment, we have added the information on the type of regressions used to the captions of Figure 1, and Table 2 (now Table 5). Figure 2 already reports that the estimates are obtained using a logistic regression model.

4. (Lines 1-2, Page 20) Authors should use caution when using the last statement in the Limitations section. As acknowledged early, no meaningful heterogeneity could be largely due to the limited sample size of the study.

We thank the reviewer for noting this point and have revised this sentence to now read:

“Our findings suggest that large differences in the main framing effects across social groups are unlikely but this is still an important consideration, especially if there are smaller differences between those that were and were not included in the study that may still be meaningful at a population level.”

Reviewer #3 (Recommendations for the authors):For future work, I would like to see a cluster analysis of participants stratified by age. To see how the framing strategies affect middle-aged adults ages 45-64 or older adults ages 65 and above.

We appreciate the reviewer’s insightful suggestion. We briefly present similar results in the “differences by country, sex, and age group” paragraph of the Results section. We do not find evidence that the effect of framing strategies on the willingness to get vaccinated varies across age; however, as mentioned in the text, we were not powered to detect small differences and will intend to follow the reviewer’s suggestion and investigate these subgroup effects in future work.